# Utilization of preconception care and associated factors in Hosanna Town, Southern Ethiopia

Meron Admasu Wegene[1]*, Negeso Gebeyehu Gejo[2], Daniel Yohannes Bedecha[2], Amene Abebe Kerbo[3], Shemsu Nuriye Hagisso[3], Solomon Abrha Damtew[3]

1 Southern Nation Nationalities Peoples Region Health Bureau, Hadiya Zone Health Department, Lemo Woreda Health Office, Hosanna, Ethiopia, 2 Department of Midwifery, School of Health Sciences, Madda Walabu University Shashemene Campus, Shashemene, Ethiopia, 3 School of Public Health, College of Medicine and Health Sciences, Wolaita Sodo University, Wolaita Sodo, Ethiopia

* merichocandy@gmail.com

**Data Availability Statement:** All relevant data are within the paper and its Supporting Information files.

**Funding:** The author(s) received no specific funding for this work.

## Abstract

### Introduction

There is substantial body of evidence that portrays gap in the existing maternal and child health continuum of care; one is less attention given to adolescent girls and young women until they get pregnant. Besides, antenatal care is too late to reduce the harmful effects that a woman's may have on the fetus during the critical period of organogenesis. Fortunately, preconception care can fill these gaps, enhance well-being of women and couples and improve subsequent pregnancy and child health outcomes. Therefore, the main aim of the current study was to assess preconception care utilization and associated factors among pregnant women attending antenatal care clinics of public health facilities in Hosanna town.

### Methods

A facility based cross-sectional study design was carried out from July 30, 2020 to August 30, 2020. Data were collected through face-to-face interview among 400 eligible pregnant women through systematic sampling technique. Epi-data version 3.1 and SPSS version 24 was used for data entry and analysis respectively. Both bivariable and multivariable logistic regression analysis was conducted to identify association between dependent and independent variables. Crude and adjusted odds ratio with respective 95% confidence intervals was computed and statistical significance was declared at p-value <0.05.

### Result

This study revealed that 76 (19%, 95% CI (15.3, 23.2) study participants had utilized preconception care. History of family planning use before the current pregnancy (AOR = 2.45; 95% CI (1.270, 4.741), previous history of adverse birth outcomes (AOR = 3.15; 95% CI (1.650, 6.005), poor knowledge on preconception care (AOR = 0.18; 95% CI (0.084, 0.379) and receiving counseling on preconception care previously (AOR = 2.82; 95% CI (1.221, 6.493) were significantly associated with preconception care utilization.

**Competing interests:** The authors have declared that no competing interests exist.

**Abbreviations:** AOR, Adjusted odds ratio; ANC, Antenatal Care; STI, Sexually Transmitted Infection; SPSS, Statistical Package for Social Science; VIF, Variance inflation factor; WHO, World Health Organization.

## Conclusions

The present study revealed that nearly one-fifth of pregnant women have utilized preconception care services. History of family planning use before the current pregnancy, previous history of adverse birth outcomes, poor knowledge on preconception care and receiving counseling on preconception care previously were significantly associated with preconception care utilization. Integrating preconception care services with other maternal neonatal child health, improving women's/couples knowledge & strengthening counseling services is pivotal.

## Introduction

Preconception care is the provision of biomedical, behavioral and social health interventions to women prior to the conception, intended to improve health status, and subsiding behavioral and environmental elements that could result in adverse maternal and child health outcomes [1]. According to the recommendation of World Health Organization, the packages of interventions for preconception care include but not limited to; maternal nutrition, cessation of tobacco and alcohol use, management of infertility and subfertility, prevention of too early, rapid and unwanted successive pregnancy and prevention and treatment of Sexually transmitted infections (STIs), Human immune-deficiency virus(HIV) counseling and treatment [2].

In 2017, globally there were 295,000 maternal deaths. Roughly, 86% of the estimated global maternal deaths is contributed by Sub-Saharan Africa and Southern Asia, while sub-Saharan Africa only contributing for nearly 66%. Similarly, 5.3 million deaths occurred in the first five years of life in 2018; of which, 2.5 million occurred in the first month of life. In Ethiopia, the maternal mortality ratio, under-five mortality rate and neonatal mortality rate is estimated at 412 per 100,000 live births,67 per 1000 live birth and 29 deaths per 1000 live births respectively [3–5].

Preconception care can make a substantial public health benefit contribution in decreasing maternal and childhood mortality and morbidity, and improving maternal and child health in both high- and low-income countries. In low-income countries, similar but larger effects may be realized through improving maternal and child health outcomes in large segments of the population [1].

Evidences portrays that existing maternal child health care continuum of care are essential to reduce maternal and childhood mortality. However, there is substantial body of evidence that shows gap in the existing continuum of care. One of the gap is that, antenatal care is too late to reduce the harmful effects that a woman's (and her partner's) health risks or health problems may have on the fetus during the decisive moment of organ formation. Preconception care completes the continuum, ensuring ongoing health surveillance and early intervention, so that women begin pregnancy in the best health possible [6].

Besides, preconception care offers the potential for earlier risk assessment and intervention that can help the woman or couple even before pregnancy and safeguard the healthiest possible beginning for their newborn child. Preconception care has also important health and social benefits. It could make a considerable role in subsiding maternal and childhood mortality and morbidity and to the health of babies as children as they grow into adolescence and adulthood. Moreover, by assisting women to make well-informed and decisive choices, preconception care could bring social and economic development for the families and society at large [7].

Preconception care in Nigeria, Sudan and Brazil was 10.3%, 9% and 15.9% respectively [8–10]. However, preconception care was relatively higher in developed countries like China, Maryland and London which was 20.6%, 33.1% and 27% [11–13].

Developed countries have strategies for preconception care at hand whereas disappointingly it's much abandoned maternal health care service that necessities urgent attention in developing nations. The intention among women and their parents to pursue preconception care is still insufficient in low income countries in spite of high pre-pregnancy risk factors [14].

In spite of preconception care is so essential to improve pregnancy and child health outcomes, relatively little is known about the pre-pregnancy health care of reproductive aged women. In Ethiopia, preconception care services are being provided in the maternal and child health care unit and other chronic disease follow up clinics. However, it is not expanded and given as per standard [15–17]. Therefore, the aim of this study is to assess preconception care utilization and factors associated with preconception care among pregnant women attending antenatal care clinics of public health facilities in Hosanna town.

## Methods

A facility based cross-sectional study design was carried out from July 30, 2020 to August 30, 2020.The study was conducted in Hosanna town, capital of Hadiya zone, Southern Nations Nationalities and People Regional State of Ethiopia. The town is situated 232 Kms southwest of Addis Ababa and 194 Kms northwest of the regional capital, Hawassa. The town has one referral hospital and three public health centers.

The source population contained all pregnant women attending antenatal care (ANC) clinics of public health institutions of Hosanna town. The source population included all systematically selected pregnant women attending ANC clinics of public health institutions of Hosanna town during the study period.

Sample size was determined using single population proportion formula by considering the following assumptions: 95% confidence level, 5% margin of error and prevalence of preconception care utilization by pregnant women which is 38.2% (taken from a research conducted in West Shoa Zone, Oromia region) [18].

There are four public health institutions in the town namely; Hosanna health center, Lichamba health center, Bobicho health center and Nigist Eleni Mohammed Memorial Referral Hospital; and all were incorporated in to this study. The monthly average number of women attending antenatal care in each public health institution was as follows; 254 in Hosanna health center, 225 in Lichamba health center, 205 in Bobicho health center and 300 in Nigist Eleni Mohammed Memorial Referral Hospital. Considering the above client flow per month in the respective health institutions, the total sample size determined (400) was distributed to the each health institutions through probability proportional to size allocation. Finally, study participants were selected by systematic sampling technique. Every $2^{nd}$ woman was taken and the first mother was selected using lottery method.

The questionnaire was prepared in English, translated to Amharic, and then translated back to English to check for consistency before commencing data collection. Training was given to data collectors and supervisors. The data collection procedure was checked for completeness and consistency on the same day by the supervisors and principal investigator. The questionnaire was pre-tested on 5% (20) of the calculated sample size in Shurmo health center. The validity of the questionnaire was tested using pears correlation and found to be valid. Reliability was also tested using Cronbach's alpha co-efficient test and found to be reliable.

Data were collected using structured and pretested questionnaire through face to face interview. The questionnaire was adapted from different related literatures to outfit the objectives of the study [18–22]. Information collected were socio-demographic characteristics, reproductive & medical characteristics, health care service related factor, knowledge and attitudes of mothers on preconception care and utilization of preconception care. Data were collected by 4 data collectors and 2 supervisors.

## Measurement

### Preconception care services

Preconception care services assessed by this study are; screening & treatment for chronic medical diseases, screening & treatment for STIs, HIV testing & counselling, diagnosis & treatment for infertility/sub-fertility, getting vaccination for tetanus, taking folic acid supplementation, follow-up and care for preexisting chronic medical condition, follow-up and care for previous adverse pregnancy & birth outcomes, weight management, diet modification, avoiding smoking & drinking alcohol, avoiding teratogenic & illicit drugs, avoiding chemical/radiation exposure in occupational, environmental and medical settings [20].

### Utilization of preconception care

If women reported receiving at least one preconception care services mentioned above either screening and treatment or follow-up and care for preexisting health problems or undertaken life style modification regarding preconception care before conceiving index pregnancy to 3 months after conceiving index pregnancy by health care providers [19].

### Non-utilization of preconception care

If women reported not receiving any of preconception care services mentioned above either screening and treatment or follow-up and care for preexisting health problems or undertaken life style modification regarding preconception care before conceiving index pregnancy to 3 months after conceiving index pregnancy by health care providers [19].

### Knowledge on preconception care

Women level of knowledge was measured using 17 questions measuring level of knowledge on preconception care. Each question has one correct answer. The score for each correct answer was one point and the score for each incorrect answer was zero point. Responses for each question was added and the total score ranged from 0–17 points. Those respondents who scored $\geq$ mean were labeled as *"adequate knowledge"* and those respondents who scored $<$ mean were labeled as *"inadequate knowledge"* [20, 21].

### Attitude towards preconception care

Women attitude towards preconception care was measured using 12 likert scale questions with five scales (strongly agree, agree, neutral, disagree, and strongly disagree). The score was (strongly agree = 5, agree = 4, neutral = 3, disagree = 2, and strongly disagree = 1) for positive statements and vice versa for negative statements and the total score ranged from 12–60 points. Responses for each question was added & divided by 12; which was the total number of questions to compute for mean. Those respondents who scored $\geq$mean&$<$mean were labeled as *"favorable attitude"* and *"unfavorable attitude"* respectively [9].

## Data analysis

All the questionnaires were checked manually for completeness and were cleaned, coded and entered in to Epi-data 3.1. Then, data was analyzed using SPSS version 24.0. Descriptive statistics were done and presented in tables and figures.

Initially, variables with p < 0.25 at bivariable logistic regression were taken in to multivariable logistic regression model. Both crude and adjusted odds ratio with respective 95% confidence intervals and p value was used to measure the strength of association between dependent and independent variables. Variables with p value <0.05 were considered as statistically significantly associated with the dependent variable.

Goodness of fit of the model was checked using Hosmer and Lemeshow test of assumption. It had a chi-square value of 8.067 & p-value of 0.327 which showed the model to be fit. Multicollinearity was checked for interaction between independent variables by VIF (Variance inflation factor). All the variables had VIF less than 5.

## Ethics approval and consent to participate

Ethical approval was taken from Wolaita Sodo University, College of medicine and health science, School of public health, Ethical Board Committee. Formal letter was obtained from Hosanna town health office. Concerned bodies from respective health facilities were officially communicated before commencing the data collection. In addition, informed written consent was obtained from study participants to confirm willingness for participation after explaining the objective of the study. For participants who were unable to write, a right thumbprint was taken as a signature. Parental or legal guardian consent was taken for respondents who were under 18 years of age. Respondents were notified about their right to refuse or terminate at any point of the interview. The information provided by each respondent was kept confidential.

## Results

### Socio-demographic characteristics

A total of 400 study participants were involved in this study with the response rate of 100%. The age of women ranges from 16 to 36. The median maternal age was 25 years (IQR 22, 28). Almost all of the study participant were married 398 (99.5%). Three hundred twenty seven (81.8%) were Hadiya and 344 (86%) were protestant. One hundred sixty five (41.3%) women had attended primary education. More than half of women were housewives 257 (64.3%). Nearly three forth of study participants earned 57.17USDper month 288 (72.0%). Almost all of the study participants resided in the urban area 373 (93.3%) (Table 1).

### Reproductive characteristics

The median gravidity and parity was 1 (IQR 1, 3) and 1 (IQR 1, 2) respectively. Nearly half of the women had never given birth 186 (46.5%). The median age at first birth was 21 (IQR 19, 24). Thirty women (7.5%) had birth interval of less than 24 months. The median duration of pregnancy was 7 months (IQR 5, 8). More than half of the women had received antenatal care two to three times 227 (56.8%).

More than one quarter of the women had ever used any family planning method 109 (27.3%). More than one fourth of the women had used modern family planning recently 116 (29.0%). More than half of the women had planned any of their previous pregnancies 230 (57.5%). Almost all, women had planned their current pregnancy 375 (93.8%) (Table 2).

**Table 1. Socio-demographic characteristics of women attending antenatal care clinics of public health facilities in Hossana town, Southern Ethiopia, 2020 (n = 400).**

| Variables | Category | Frequency | % |
|---|---|---|---|
| Age | 15–19 | 25 | 6.3 |
| | 20–24 | 135 | 33.8 |
| | 25–29 | 189 | 47.3 |
| | 30–34 | 38 | 9.5 |
| | > = 35 | 13 | 3.3 |
| | Total | 400 | 100.0 |
| Marital status | Married | 398 | 99.5 |
| | Single | 1 | 0.25 |
| | Widowed | 1 | 0.25 |
| | Total | 400 | 100.0 |
| Religion | Orthodox | 32 | 8.0 |
| | Muslim | 15 | 3.8 |
| | Protestant | 344 | 86.0 |
| | Catholic | 4 | 1.0 |
| | Adventist | 5 | 1.3 |
| | Total | 400 | 100.0 |
| Ethnicity | Hadiya | 327 | 81.8 |
| | Kembata | 17 | 4. 3 |
| | Silte | 9 | 2. 3 |
| | Gurage | 19 | 4.8 |
| | Amhara | 11 | 2.8 |
| | Oromo | 10 | 2.5 |
| | Woliata | 7 | 1.8 |
| | Total | 400 | 100.0 |
| Educational status of mothers | No formal education | 11 | 2.8 |
| | Read and write | 23 | 5.8 |
| | Primary education | 165 | 41.3 |
| | Secondary education | 104 | 26.0 |
| | Diploma & above | 97 | 24.3 |
| | Total | 400 | 100.0 |
| Monthly income | <14.28USD* | 7 | 1.8 |
| | 14.31–28.57USD | 58 | 14.5 |
| | 28.6–57.14USD | 47 | 11.8 |
| | > = 57.17USD | 288 | 72.0 |
| | Total | 400 | 100.0 |
| Residence | Urban | 373 | 93.3 |
| | Rural | 27 | 6.8 |
| | Total | 400 | 100.0 |

*1ETB = 0.02857USD

## Obstetrics and medical characteristics

About 18 (4.5%) women had experienced complications during pregnancy. The most commonly experienced complications was antepartum hemorrhage 13 (3.3%). About 56 (14.0%) had experienced adverse pregnancy outcomes. The most commonly experienced adverse pregnancy outcomes were abortion 39 (9.8%) and stillbirth 9 (2.3%). About nine (2.2%) women

**Table 2. Reproductive characteristics of women attending antenatal care clinics of public health facilities in Hossana town, Southern Ethiopia, 2020 (n = 400).**

| Variables | Category | Frequency | % |
|---|---|---|---|
| Gravidity | 1 | 168 | 42.0 |
| | 2–4 | 217 | 54.2 |
| | 5+ | 15 | 3.8 |
| | Total | 400 | 100.0 |
| Gave birth | Yes | 214 | 53.5 |
| | No | 186 | 46.5 |
| | Total | 400 | 100.0 |
| Parity | 1 | 121 | 30.3 |
| | 2–3 | 82 | 20.5 |
| | 4–5 | 10 | 2.5 |
| | 6+ | 1 | 0.2 |
| | Primigravida | 186 | 46.5 |
| | Total | 400 | 100.0 |
| Birth interval | <24 | 30 | 7.5 |
| | 25–47 | 49 | 12.3 |
| | >48 | 11 | 2.7 |
| | One Child | 124 | 31.0 |
| | Primigravida | 186 | 46.5 |
| | Total | 400 | 100.0 |
| Pregnancy duration | <4 | 46 | 11.5 |
| | 5–6 | 153 | 38.3 |
| | 7–8 | 121 | 30.3 |
| | 9+ | 80 | 20.0 |
| | Total | 400 | 100.0 |
| Frequency of antenatal care | 1 | 116 | 29.0 |
| | 2–3 | 227 | 56.8 |
| | 4–5 | 57 | 14. 3 |
| | Total | 400 | 100.0 |
| Ever used family planning | Yes | 109 | 27. 3 |
| | No | 291 | 72.7 |
| | Total | 400 | 100.0 |
| Recent type of family planning used | Intrauterine contraceptive device | 4 | 1.0 |
| | Injectable | 33 | 8.3 |
| | Implants | 65 | 16.3 |
| | Pill | 14 | 3.4 |
| | Not using family planning recently | 284 | 71.0 |
| | Total | 400 | 100.0 |
| Planned any of previous pregnancies | Yes | 230 | 57.5 |
| | No | 4 | 1.0 |
| | Primigravida | 186 | 46.5 |
| | Total | 400 | 100.0 |
| Planned current pregnancy | Yes | 375 | 93.8 |
| | No | 25 | 6.2 |
| | Total | 400 | 100.0 |

had medically confirmed disease and the most commonly confirmed medical disease was chronic hypertension 3 (0.8%). About 10 (2.5%) women had family history of medically

confirmed disease and the most common family history of confirmed medical diseases were chronic hypertension 6 (1.5%) and diabetes mellitus 3 (0.8%) (Table 3).

## Health service related factors

Majority of women resided within 5kms of health facility 379 (94.8%) and almost all women had no challenge in accessing health facility 394 (98.5%).

**Table 3. Obstetrics and medical characteristics of women attending antenatal care clinics of public health facilities in Hossana town, Southern Ethiopia, 2020.**

| Variables | Category | Frequency | % |
|---|---|---|---|
| Experienced pregnancy related complication previously | Yes | 18 | 4.5 |
| | No | 196 | 49.0 |
| | Primigravida | 186 | 46.5 |
| | Total | 400 | 100.0 |
| Type of pregnancy related complications | Antepartum hemorrhage | 13 | 3.3 |
| | Preeclampsia | 2 | 0.5 |
| | Rh incompatibility | 2 | 0.5 |
| | Preeclampsia & Gestational DM | 1 | 0.2 |
| | Not developed pregnancy related complications | 196 | 49.0 |
| | Primigravida | 186 | 46.5 |
| | Total | 400 | 100.0 |
| Experienced adverse birth outcomes | Yes | 56 | 14.0 |
| | No | 158 | 39.5 |
| | Primigravida | 186 | 46.5 |
| | Total | 400 | 100.0 |
| Types of adverse birth outcomes experienced | Preterm neonate | 6 | 1.5 |
| | Abortion | 39 | 9.8 |
| | Stillbirth | 9 | 2.3 |
| | Neonatal death | 1 | 0.2 |
| | Congenital anomalies & abortion | 1 | 0.2 |
| | Not experienced adverse birth outcomes | 158 | 39.5 |
| | Primigravida | 186 | 46.5 |
| | Total | 400 | 100.0 |
| Medically confirmed disease | Yes | 9 | 2.2 |
| | No | 391 | 97.8 |
| | Total | 400 | 100.0 |
| Types of medically confirmed diseases | Diabetes mellitus | 1 | 0.2 |
| | Chronic hypertension | 3 | 0.8 |
| | Asthma | 2 | 0.5 |
| | Cardiac diseases | 1 | 0.2 |
| | Anemia | 2 | 0.5 |
| | No history of medically confirmed disease | 391 | 97.8 |
| | Total | 400 | 100.0 |
| Family history of medically confirmed diseases | Yes | 10 | 2.5 |
| | No | 390 | 97.5 |
| | Total | 400 | 100.0 |
| Types of family history medically confirmed diseases | Diabetes mellitus | 3 | 0.8 |
| | Chronic hypertension | 6 | 1.5 |
| | Asthma | 1 | 0.2 |
| | No family history of medically confirmed disease | 390 | 97.5 |
| | Total | 400 | 100.0 |

Thirty four (8.5%) women had received counseling on preconception care in previous pregnancies and commonly received counseling were diet modification before conception 14 (41.2%) and counseling on folic acid supplementation 6 (17.6%) (Fig 1).

## Knowledge on preconception care

More than half of study participants knew that women health and lifestyle before pregnancy can influence both the fertility and the health of mother, and child 246 (61.5%). 56 (14%) study participants didn't know about women preparation and maintenance of health before getting pregnant. 117 (29.3%) women have heard about preconception care and they got the information mostly from health professionals including health extension workers 43 (36.8%) and mass media 35 (29.9%).

More than half of the study participants didn't know the things that should be done before pregnancy 218 (54.5%). Out of women who knew the things that should be done before pregnancy, majority of them mentioned visiting health facility to seek advice on healthy pregnancy 35 (19.2%) and less commonly they mentioned quitting smoking and drinking alcohol 1 (0.5%), and avoiding teratogenic and illicit drugs 1 (0.5%).

One hundred seventy (42.5%) respondents didn't know chronic medical conditions that affect the fetus. From those respondents who knew chronic medical conditions that affect the fetus; 69 (17.3%) mentioned, STIs including HIV while 39 (9.8%) women mentioned, DM and

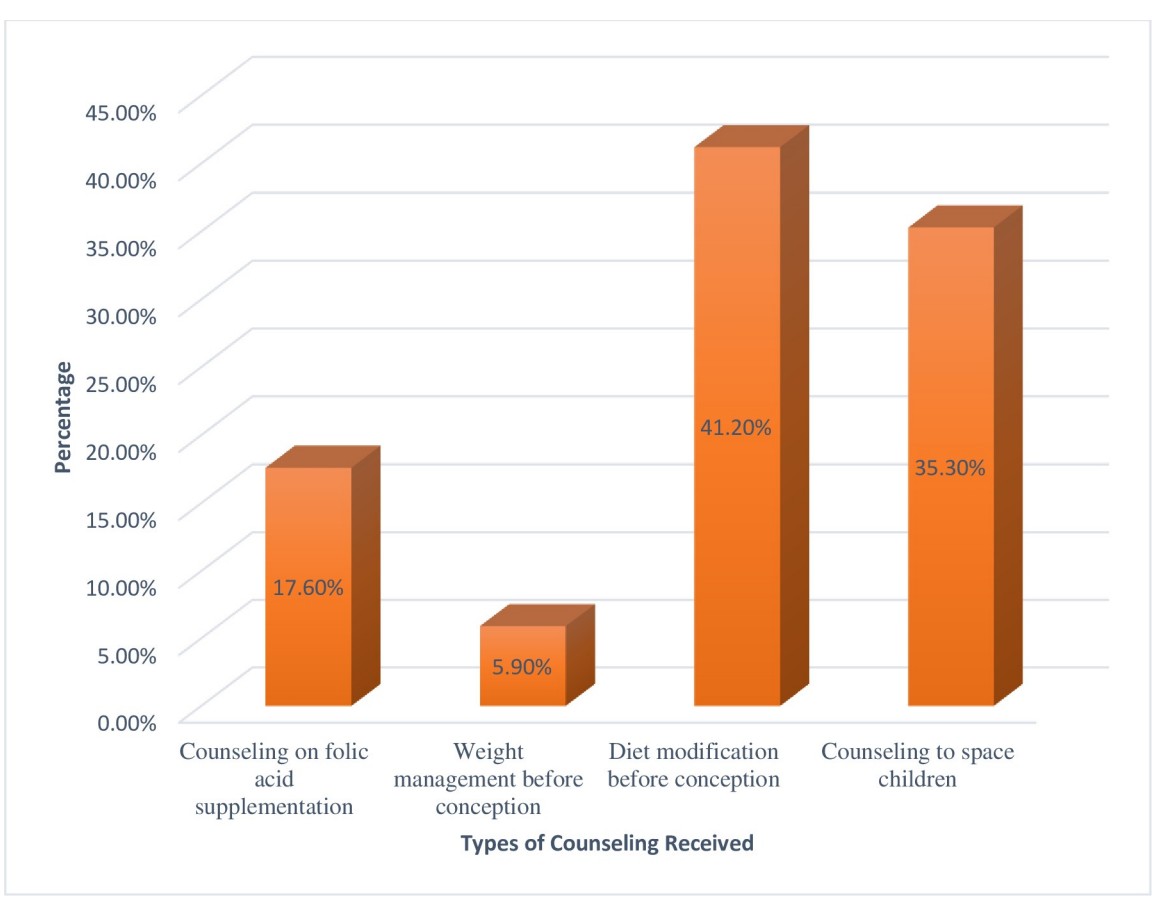

**Fig 1. Types of counseling received on preconception care among women attending antenatal care clinics of public health facilities in Hossana town, Southern Ethiopia, 2020.**

chronic hypertension. More than one third of the women didn't know lifestyle/ behavioral/ environmental factors that affect the fetus.

More than half of study participants thought that preconception care is important for all women 256 (64.0%) and about 73 (18.3%) women didn't know for whom preconception care is important. One hundred forty respondents (35.8%) didn't know the time couples/women receive preconception care services.

More than one third of study participants didn't know how frequently preconception care services need to be provided 149 (37.3%) while 173 (43.3%) women stated that preconception care services need to be provided continuously.

Regarding overall knowledge of the study participants, more than half of the respondents had adequate knowledge on preconception care 259 (64.8%).

## Attitude towards preconception care

One hundred seventy five respondents (43.85) strongly agreed on preconception care to be a high health care priority for all women/couples planning pregnancy. About 66 (16.5%) respondents strongly agreed that women with medically confirmed diseases should only receive preconception care services. Two hundred eighty seven (71.8%) women disagreed on the point that preconception care should be provided for unmarried women.

About 183 (45.8%) of the study participants strongly agreed that husbands should accompany their wives while seeking any preconception care services. Roughly 293 (73.3%) women disagreed on the point that preconception care has no effect on the birth outcome. Regarding the overall attitude of women towards preconception care, about 59 (14.7%) respondents had unfavorable attitude.

## Preconception care utilization

Among women who were pregnant previously, about 17 (4.3%) received preconception care services from health facility during any of their previous pregnancies. The most commonly received preconception care services were being screened and treated for chronic medical diseases and HIV counseling & testing 5 (29.4%).

Regarding current utilization of preconception care, 76 (19%, 95% Cl 15.3, 23.2) study participants had utilized preconception care (Fig 2). The most commonly utilized preconception care were HIV counseling and testing exclusively 15 (19.7%) and being screened and treated for chronic medical diseases exclusively 14 (18.4%). Majority of the women received preconception care when they plan to get pregnant 49 (64.5%) while 9 (11.8%) received during the first three months of pregnancy.

More than half of the respondents received preconception care from public hospitals 42 (55.3%) and mostly from adult OPD 40 (52.6%). About 61 (80.3%) had received preconception care once. Almost all of the study participants made decision on pregnancy together (husbands and wives) 74 (55.3%) & all of the husbands support and accompany their wives while seeking preconception care services.

Among 46 women who were screened for chronic medical conditions (14 exclusively screened for chronic medical diseases and 32 screened for chronic medical diseases plus received other preconception care services); nearly half of them were screened for DM and hypertension 22 (47.8%) while 19 (41.5%) women were screened for hypertension alone. Among respondents who managed their weight as a preconception care, three women lost their weight while two gained weight. Among the women who took folic acid as a preconception care, all of them had taken it consistently. Among those who modify their diet as a preconception care, seven had eaten balanced diet while two avoided foods with high saturated fat.

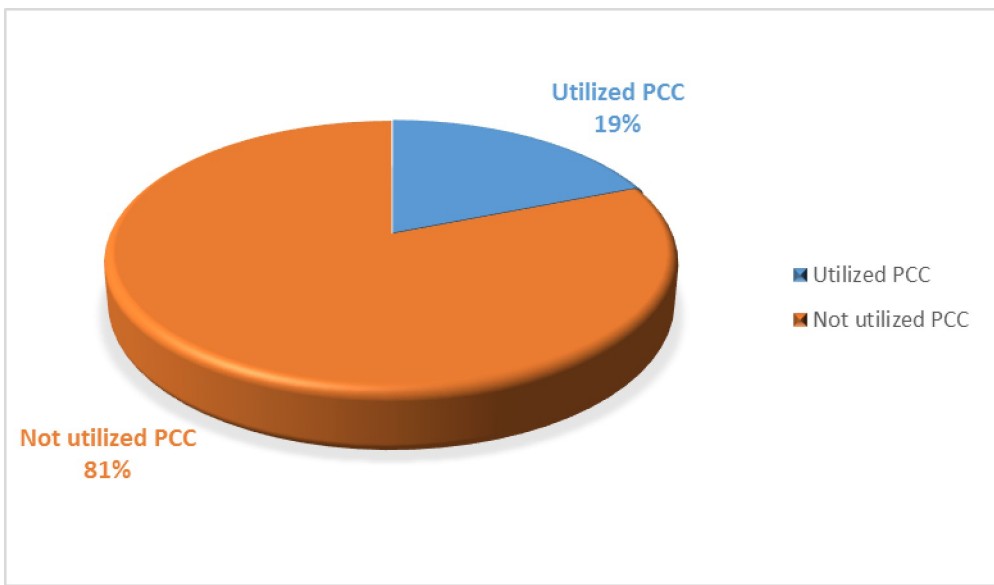

**Fig 2. Utilization of preconception care among women attending antenatal care clinics of public health facilities in Hossana town, Southern Ethiopia, 2020.**

### Factors associated with preconception care

Women's education, place of residence, parity, family planning use before current pregnancy, previous history of adverse birth outcome, challenge in accessing health facility, received counseling on preconception care previously, overall knowledge on preconception care and overall attitude towards preconception care had a p-value of ≤ 0.25 in the bivariable analysis and taken in to the final model for multivariable analysis (Table 4).

These variables were candidate for multivariable analysis and taken to final model to control potential confounding effect. In multivariable logistic regression analysis; family planning use before current pregnancy, previous history of adverse birth outcome, receiving counseling on preconception care previously and knowledge on preconception care were statistically significantly associated with preconception care at p-value < 0.05 (Table 4).

The odds of utilizing preconception care in a women who had used family planning before current pregnancy were 2.45 times higher than those women who had not used family planning before current pregnancy(AOR = 2.45; 95% Cl (1.270, 4.741).

The odds of utilizing preconception care in a women who had history of adverse birth outcome were 3.15 times higher than their counterpart (AOR = 3.15; 95% Cl (1.650, 6.005).

The odds of utilizing preconception care in a women who had received counseling on preconception previously were 2.82 times higher than those women who had not received counseling on preconception care (AOR = 2.82; 95% Cl (1.221, 6.493).

There is 82% reduced odds of utilizing preconception care among women who had poor knowledge on preconception care than those women with good knowledge on preconception care (AOR = 0.18; 95% Cl (0.084, 0.379).

## Discussion

The current study assessed preconception care and associated factors among women attending antenatal care clinics of public health facilities in Hossana town. In the present study, 76

**Table 4. Bivariable and multivariable logistic regression analyses for factors associated with preconception care among women attending antenatal care clinics of public health facilities in Hossana town, Southern Ethiopia, 2020.**

| Variables | Category | Preconception care utilization | | COR(95% CI) | P-value | AOR (95%) | P-value |
|---|---|---|---|---|---|---|---|
| | | Yes | No | | | | |
| | | Freq. (%) | Freq. (%) | | | | |
| Women's education | No formal education | 1 (9.1) | 10 (90.9) | 0.36 (0.107, 1.225) | 0.102 | 1.38 (0.066, 28.562) | 0.824 |
| | Read & write | 5 (21.7) | 18 (78.3) | 1.0 (0.532, 1.900) | 0.987 | 2.39 (0.652, 8.771) | 0.189 |
| | Primary education | 25 (15.2) | 140 (84.8%) | 0.65 (0.446, 0.937) | 0.021* | 0.92 (0.445, 1.844) | 0.812 |
| | Secondary education | 24 (23.1) | 80 (76.9) | 1.09 (0.740, 1.593) | 0.674 | 1.66 (0.806, 3.416) | 0.170 |
| | Diploma & more | 21 (21.6) | 76 (78.4) | 1 | 1 | 1 | 1 |
| Place of residence | Urban | 68 (18.2) | 305 (81.8) | 0.53 (0.321, 0.873) | 0.013* | 0.4 (0.145, 1.085) | 0.072 |
| | Rural | 8 (29.6) | 19 (70.4) | 1 | 1 | 1 | 1 |
| Parity | Multiparous | 37 (17.2) | 178 (82.8) | 0.78 (0.472, 1.283) | 0. 326* | 1.01 (0.403, 2.535) | 0.982 |
| | Nulliparous | 39 (21.1) | 146 (78.9) | 1 | 1 | 1 | 1 |
| Family planning use before current pregnancy | Yes | 27 (23.3) | 89 (76.7) | 1.45(1.072, 1.975) | 0.016* | **2.45 (1.270, 4.741)** | **0.008**** |
| | No | 49 (17.3) | 235 (82.7) | 1 | 1 | 1 | 1 |
| Previous history of adverse birth outcome | Yes | 17 (30.4) | 39 (69.6) | 2.63 (1.295, 5.353) | 0.008* | **3.15 (1.650, 6.005)** | **0.001**** |
| | No | 25 (14.2) | 151 (85.8) | 1 | 1 | 1 | 1 |
| Challenge in accessing health facility | Yes | 4 (36.4) | 7 (63.6) | 2.52 (1.219, 5.192) | 0.013* | 1.47 (0.342, 6.313) | 0.610 |
| | No | 72 (18.5) | 317 (81.5) | 1 | 1 | 1 | 1 |
| Counselling received on preconception care previously | Yes | 13 (38.2) | 21 (61.8) | 2.98 (1.939, 4.572) | 0.000* | **2.82 (1.221, 6.493)** | **0.015**** |
| | No | 63 (17.2) | 303 (82.8) | 1 | 1 | 1 | 1 |
| Knowledge on preconception care | Poor knowledge | 9 (6.4) | 132 (93.6) | 0.19 (0.128, 0.298) | 0.000* | **0.18 (0.084, 0.379)** | **0.000**** |
| | Good knowledge | 67 (25.9) | 192 (74.1) | 1 | 1 | 1 | 1 |
| Attitude towards preconception care | Unfavorable attitude | 6 (10.2) | 53 (89.8) | 0.44 (0.263, 0.730) | 0.002* | 0.91 (0.335, 2.458) | 0.849 |
| | Favorable attitude | 70 (20.5) | 271 (79.5) | 1 | 1 | 1 | 1 |

*Variables statistically significant at P value ≤0.25 in bivariable analyses.

** Variables statistically significant at p-value <0.05 in multivariable analyses.

(19%), 95% Cl (15.3, 23.2) study participants had utilized preconception care which coincides with the study done in Mekelle City, North Ethiopia (18.2%) [19].

This finding is slightly higher than studies carried out in Debre Birhan (13.4%), West Shoa zone; Oromia region (14.5%) and Adet town (9.6%) [20–22]. The discrepancy might be due to the difference in study design, study setting and educational status of women. For instance the studies conducted in Debre Birhan town, West Shoa zone; Oromia region, and Adet town all were community based studies conducted on reproductive age women. Besides, about 14.6% of women Debre Birhan town had no formal education whereas in the present study only 2.8% had no formal education. The more women are educated, the more likely to utilize health care services including preconception care.

However it's lower than study conducted in health centers found in West Shoa zone; Oromia region (38.2%) [18]. The discrepancy could be due to larger sample size included and dissimilarity in study setting.

The finding of the present study is also lower than studies done in Sri Lanka (27.2%) and Bachok (45.2%) [23, 24]. The difference might be due to the dissimilarity in study setting, socio-demographic characteristics of the study populations and health care policy and health care structure in the respective countries.

Family planning use before current pregnancy had statistically significant association with preconception care utilization. The odds of utilizing preconception care in a women who had

used family planning before current pregnancy were 2.45 times higher than those women who had not used family planning before current pregnancy.

This might due to women who get family planning service are alongside counseled on different reproductive issues and also when they stop using the method or removed it in the case of implants to get pregnant, they have a good opportunity to utilize several preconception care services. A cross-sectional study done in West Shoa zone; Oromia region revealed that having a history of family planning use is significantly associated with knowledge of PCC [21].

The present study is in contrary with the cross-sectional study done in Debre Birhan town where history of contraception had no statistically significant association with preconception care utilization [20]. This discrepancy might be due to difference in proportion of women who had utilized preconception care out of those who had history of family planning where only 16% of women had utilized preconception out of two hundred seventy five women having had family planning use. But in the present study, 24% of respondents had utilized preconception care out of one hundred sixteen women having had family planning use.

This implies that integrating preconception care services with other maternal neonatal child health like family planning is so important in improving and advancing preconception care utilization. This implication is also supported by World Health Organization which recommended that the selected preconception care interventions could be creatively included in delivery mechanisms that are currently being used to reach specific groups in the population [1].

Mothers who had experienced adverse birth outcomes showed statistically significant association with preconception care utilization. The odds of utilizing preconception care in a women who had experienced adverse birth outcome were 3.15 times higher than their counterpart. This might be due to the fact that mothers who faced such complications are more vigilant and tend to seek preconception care to minimize or avoid adverse birth outcomes in the subsequent pregnancies.

This finding is in line with cross-sectional study done in Mekelle city, Northern Ethiopia which revealed that women who had chronic health problems were more likely to utilize PCC compared to their counterparts [19]. It is also in line with study carried out in Los Angeles. In the Los Angeles study, having had adverse infant outcomes like preterm delivery, low birth weight, stillbirth or major birth defect was associated with an increased odds of having utilized preconception care in the most recent pregnancies [25].

This implies that especial emphasis should be given to such mothers and they need to get appropriate medical treatment and psychological support to minimize or avoid the risk of adverse birth outcomes in their subsequent pregnancies.

Knowledge on preconception care was the other factor associated with preconception care. There is 82% reduced odds of utilizing preconception care among women who had poor knowledge on preconception care than those women with good knowledge on preconception. More than one-third of respondents had poor knowledge on preconception care 141 (35.2%).

This might be due to the fact that women who had poor knowledge would not utilize preconception care. Women who had poor knowledge do not have deep understanding of preconception care and also the information they had regarding components and benefits of preconception care is very limited. The present finding is in line with the cross-sectional studies done in Mekelle city, Debre Birhan and West Shoa zone; Oromia region [19–21]. This finding is also supported by cross-sectional study done in China [11].

This implies that improving women's knowledge on the benefits of preconception care and its components is so pivotal in increasing preconception care utilization. Disseminating information regarding preconception care at schools, home and market places through health extension workers can bring a significant change.

The other factor which showed statistically significant association with preconception care was counselling received on preconception care previously. The odds of utilizing preconception care in a women who had received counselling on preconception care previously were 3.15 times higher than their counterpart.

This could explained as follows; women who received advice regarding preconception care are more likely to get preconception care and tend to undergo lifestyle modifications related with preconception care. Those women who counseled to control preexisting medical conditions or take folic acid supplementation or undergo weight management or diet modification are more likely to undertake these interventions than other women who didn't get this counseling.

This finding is supported by descriptive cross-sectional study carried out at teaching hospital in Southeast Nigeria which revealed that there were statistically significant correlation between utilization of preconception care and information from doctors [8]. It's also in line with Cross-sectional study done among pregnant women attending maternity services in London which found that women who get counseling from a health professional prior to conception were more likely to adopt healthier behaviors before taking folic acid and or adopting a healthier diet before pregnancy in London [13].

This implies that strengthening counseling services provided for women can increase women utilization of preconception services and help them to start their pregnancy at best possible health. In case if women have medically confirmed disease or history of adverse pregnancy or birth outcomes, there will be increased chance of getting treatment for these complications.

As a limitation, the current study is liable to recall bias as women were enquired about preconception care prior to the current pregnancy. Besides, husbands were not incorporated in to the current study. Moreover, since the current study is a cross-sectional survey, temporal relationship can't be clearly established.

## Conclusions

The present study revealed that nearly one-fifth of pregnant women have utilized preconception care services which calls for substantial intervention to improve the service coverage. Family planning use before current pregnancy, experiencing adverse birth outcomes, knowledge on preconception care and counselling received on preconception care previously were factors significantly association with preconception care utilization.

## Supporting information

**S1 File. PCC data set.**
(XLSX)

**S2 File. English questionnaire.**
(DOCX)

**S3 File. Amharic questionnaire.**
(DOCX)

## Acknowledgments

We are sincerely grateful to Wolaita Sodo University, officials' of the respective health facilities, data collectors, supervisors and respondents without whom the present study wouldn't be realized.

## Author Contributions

**Conceptualization:** Meron Admasu Wegene.

**Data curation:** Meron Admasu Wegene, Amene Abebe Kerbo, Shemsu Nuriye Hagisso.

**Formal analysis:** Meron Admasu Wegene, Negeso Gebeyehu Gejo, Amene Abebe Kerbo, Solomon Abrha Damtew.

**Funding acquisition:** Meron Admasu Wegene.

**Investigation:** Meron Admasu Wegene, Daniel Yohannes Bedecha, Amene Abebe Kerbo, Shemsu Nuriye Hagisso, Solomon Abrha Damtew.

**Methodology:** Meron Admasu Wegene, Amene Abebe Kerbo, Shemsu Nuriye Hagisso, Solomon Abrha Damtew.

**Project administration:** Meron Admasu Wegene.

**Resources:** Meron Admasu Wegene.

**Software:** Meron Admasu Wegene, Negeso Gebeyehu Gejo, Daniel Yohannes Bedecha, Solomon Abrha Damtew.

**Supervision:** Amene Abebe Kerbo, Shemsu Nuriye Hagisso, Solomon Abrha Damtew.

**Validation:** Meron Admasu Wegene, Daniel Yohannes Bedecha, Amene Abebe Kerbo, Solomon Abrha Damtew.

**Visualization:** Meron Admasu Wegene, Negeso Gebeyehu Gejo.

**Writing – original draft:** Meron Admasu Wegene, Negeso Gebeyehu Gejo.

**Writing – review & editing:** Negeso Gebeyehu Gejo.

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
