## [Decision Letter · Decision Letter 0]

4 Jul 2021

PONE-D-21-10971

Utilization of preconception care and associated factors in Hosanna Town, Southern Ethiopia

PLOS ONE

Dear Dr. Wegene,

Thank you for submitting your manuscript to PLOS ONE. After careful consideration, we feel that it has merit but does not fully meet PLOS ONE’s publication criteria as it currently stands. Therefore, we invite you to submit a revised version of the manuscript that addresses the points raised during the review process.

We look forward to receiving your revised manuscript.

Kind regards,

Nülüfer Erbil, Ph.D, Prof.

Academic Editor

PLOS ONE

Journal Requirements:

2. Please include your actual numerical p-values in Table 4.

3. In your methods, please provide the names of the four public health institutions in your study.

5. We note you have included a table to which you do not refer in the text of your manuscript. Please ensure that you refer to Table 4 in your text; if accepted, production will need this reference to link the reader to the Table.

Reviewers' comments:

Reviewer's Responses to Questions

**Comments to the Author**

1. Is the manuscript technically sound, and do the data support the conclusions?

Reviewer #1: No

Reviewer #2: No

2. Has the statistical analysis been performed appropriately and rigorously? 

Reviewer #1: No

Reviewer #2: Yes

3. Have the authors made all data underlying the findings in their manuscript fully available?

Reviewer #1: No

Reviewer #2: Yes

4. Is the manuscript presented in an intelligible fashion and written in standard English?

Reviewer #1: Yes

Reviewer #2: Yes

5. Review Comments to the Author

Reviewer #1: Dear Author,

This paper aims use a cross-sectional to investigate the Utilization of preconception care and associated factors in Hosanna Town, Southern Ethiopia. It’s interesting, however, my overall impression of this work is that it is not ready for publication owing to the following reasons:

This article is quite an interesting research, but could not clearly see each Questionnaire's relevant references in the introduction section. for example, Measurement section, p136~167, for more contribution, it’s would be better-cited references. In addition, suggest authors statistic analysis by professional proof.

-Abstract Result section, please check related utilization of preconception care data, page 41~46, page 284~289, and Page 332~334 table 4, because the authors’ table presents that COR(95% CI) is incorrect.

-Page 196~198 : “Three hundred twenty seven 197 (81.8%) were Hadiya….” And “More than half of women were housewives 257 (64.3%). Nearly…”Please check, I could not find them in table 1.

-Page 204~206: Reproductive characteristics section, “Nearly half of 206 the women had never given birth 186 (46.5%). The median age at first birth was 21 (IQR 19, 24). 207 Out of women who had 2 or more children, about one third had birth…” and “More than half 209 of the women had received antenatal care two to three times 227 (56.8%)….”Please check, I could not find those data in table 2.

-Page 210~212: please serious care your data, “More than one quarter of the women had ever used any family planning method 109 (27.3%). More than one fourth of the women had used modern family planning before current pregnancy 116 (29.0%).” Please check, I could not got them in table 2.

-Page 234~236, please check table 2, where authors miss some data, such usthe total samples 400, but Parity total number only 214, Birth interval total number only 90, Type of family planning used total number only 116, Planned any of previous pregnancies total number only 234, please carefully deal with them.

-table 3 and table 4, all data had the same problem with table2, please carefully deal with them. So authors should be rewording Obstetrics and medical characteristics section base on incorrect table 3.

-Page 225~230: Health service-related factors section, authors should rewording, because figure 4 could not find “Majority of women resided within 5kms of health facility 379 (94.8%)and almost all women had 227 no challenge in accessing health facility 394 (98.5%). Thirty-four (8.5%) women had received counseling on preconception care in previous pregnancies…” please carefully deal with data.

-P242~267, Knowledge on preconception care section, base on the study’s total samples 400, why some missing data not present in table, pleases rewording the part base on your data.

-P268~278, Attitude towards preconception care section, the same question with the incorrect tables.

-P279~302, Preconception care utilization section, the same question with the incorrect tables.

- P303~324, Factor sassociated with preconception care section, the same question with theincorrect tables.

-P337~428, Discussion section, the same question with the incorrect tables.

-The limitation part, so few, it’s should add more related the study.

-In the discussion section, suggest you add your finding linked with the study’s hypothesis.

- Some references incorrect, authors should follow PLoS one guidelines style.

Thank you for your efforts.

Reviewer #2: Thank you for offering the opportunity to referee this article.

The article overall is well written, I just have a few small suggestions.

In the introduction, it should be emphasized better why it is important of preconseption care utulization for the region where you work.

In the results, the findings are given in detail in the tables, the text part can be given more summary.

What are the limitations specific to the study should be added to the limitations section.

6. PLOS authors have the option to publish the peer review history of their article (what does this mean?). If published, this will include your full peer review and any attached files.

Reviewer #1: No

Reviewer #2: No

---

## [Author Response · Author response to Decision Letter 0]

17 Jul 2021

We have revised the manuscript as per comment given. So, we will kindly invite reviewers to check the revised manuscript. Thank you for your time.

---

## [Decision Letter · Decision Letter 1]

3 Dec 2021

PONE-D-21-10971R1Utilization of preconception care and associated factors in Hosanna Town, Southern EthiopiaPLOS ONE

Dear Dr. Wegene,

Thank you for submitting your manuscript to PLOS ONE. After careful consideration, we feel that it has merit but does not fully meet PLOS ONE’s publication criteria as it currently stands. Therefore, we invite you to submit a revised version of the manuscript that addresses the points raised during the review process.

We look forward to receiving your revised manuscript.

Kind regards,

Marianne Clemence, Associate Editor, PLOS ONE, on behalf of,

Nülüfer Erbil

Academic Editor

PLOS ONE

Journal Requirements:

Please note that Reviewer 3 comments have been included as an attachment.

Additional Editor Comments (if provided):

Reviewers' comments:

Reviewer's Responses to Questions

**Comments to the Author**

1. If the authors have adequately addressed your comments raised in a previous round of review and you feel that this manuscript is now acceptable for publication, you may indicate that here to bypass the “Comments to the Author” section, enter your conflict of interest statement in the “Confidential to Editor” section, and submit your "Accept" recommendation.

Reviewer #1: (No Response)

Reviewer #2: All comments have been addressed

Reviewer #3: (No Response)

2. Is the manuscript technically sound, and do the data support the conclusions?

Reviewer #1: No

Reviewer #2: Yes

Reviewer #3: (No Response)

3. Has the statistical analysis been performed appropriately and rigorously? 

Reviewer #1: No

Reviewer #2: Yes

Reviewer #3: Yes

4. Have the authors made all data underlying the findings in their manuscript fully available?

Reviewer #1: No

Reviewer #2: Yes

Reviewer #3: (No Response)

5. Is the manuscript presented in an intelligible fashion and written in standard English?

Reviewer #1: Yes

Reviewer #2: Yes

Reviewer #3: (No Response)

6. Review Comments to the Author

Reviewer #1: This revised article still has some section need to clarify, Questionnaire's relevant references are not presented from lines 136-167.

. In addition, the author’s statistical analysis stated that it performed appropriately. But table2, table 3, and table 4 still reported incorrect data; Statistical information is incorrect, easy to mislead readers. Please follow PLOS ONE guidelines for statistical reporting and read more about PLOS ONE paper on how to correct reporting.

https://plos.org/resource/how-to-report-statistics/

Shabrawishi, M., Al-Gethamy, M. M., Naser, A. Y., Ghazawi, M. A., Alsharif, G. F., Obaid, E. F., ... & Alwafi, H. (2020). Clinical, radiological, and therapeutic characteristics of patients with COVID-19 in Saudi Arabia. PLoS One, 15(8), e0237130.

Azlan, A. A., Hamzah, M. R., Sern, T. J., Ayub, S. H., & Mohamad, E. (2020). Public knowledge, attitudes, and practices towards COVID-19: A cross-sectional study in Malaysia. PloS one, 15(5), e0233668.

Moreover, all content DATA should check base on revised data.

Reviewer #2: (No Response)

Reviewer #3: (No Response)

7. PLOS authors have the option to publish the peer review history of their article (what does this mean?). If published, this will include your full peer review and any attached files.

Reviewer #1: No

Reviewer #2: No

Reviewer #3: No

---

## [Author Response · Author response to Decision Letter 1]

8 Dec 2021

We would to thank all reviewers for their constructive comments. Specifically, we addressed comments given by reviewer 1. We kindly request him/her to evaluate the corrected manuscript.

---

## [Editor Report · Decision Letter 2]

14 Dec 2021

Utilization of preconception care and associated factors in Hosanna Town, Southern Ethiopia

PONE-D-21-10971R2

Dear Dr. Wegene,

We’re pleased to inform you that your manuscript has been judged scientifically suitable for publication and will be formally accepted for publication once it meets all outstanding technical requirements.

Kind regards,

Nülüfer Erbil, Ph.D, Prof.

Academic Editor

PLOS ONE
---

## [Editor Report · Acceptance letter]

31 Dec 2021

PONE-D-21-10971R2 

Utilization of preconception care and associated factors in Hosanna Town, Southern Ethiopia 

Dear Dr. Wegene:

I'm pleased to inform you that your manuscript has been deemed suitable for publication in PLOS ONE. Congratulations! Your manuscript is now with our production department. 

Kind regards, 

on behalf of

Dr. Nülüfer Erbil 

Academic Editor

PLOS ONE